# Deleuze Becoming-Mary Poppins: Re-Imagining the Concept of Becoming-Woman and Its Potential for Challenging Current Notions of Parenting, Gender and Childhood

**Donna Carlyle** [1,*] and **Kay Sidebottom** [2]

1  Department of Social Work, Education and Community Wellbeing, University of Northumbria, Newcastle upon Tyne NE7 7XA, UK
2  Carnegie School of Education, Leeds Beckett University, Leeds LS1 3HE, UK; K.Sidebottom@leedsbeckett.ac.uk
*  Correspondence: donna.carlyle@northumbria.ac.uk

**Abstract:** In this paper, we consider the major and controversial lexicon of Deleuze's 'becoming-woman' and what an alternative re-working of this concept might look like through the story of Mary Poppins. In playfully exploring the many interesting aspects of Travers' character, with her classic tale about the vagaries of parenting, we attempt to highlight how reading Mary Poppins through the Deleuzian lens of 'becoming-woman' opens up possibilities, not limitations, in terms of feminist perspectives. In initially resisting the 'Disneyfication' of Mary Poppins, Travers offered insights and opportunities which we revisit and consider in terms of how this fictional character can significantly disrupt ideas of gender performativity. We endeavour to accentuate how one of its themes not only dismantles the patriarchy in 1910 but also has significant traction in the twenty- first century. We also put forth the idea of Mary Poppins as an icon of post-humanism, a nomadic war machine, with her robotic caring, magic powers and literal flights of fancy, to argue how she ironically holds the dual position of representing the professionalisation of parenting and the need to move beyond a Dionysian view of children as in need of control and regulation, as well as that of nurturer and emancipator. Indeed, in her many contradictions, we suggest a nomadic Mary Poppins can offer a route into the ideas of Deleuze and his view of children as de-territorialising forces and activators of change.

**Keywords:** Deleuze's philosophy; childhood; becoming-woman; Mary Poppins; parenting; child-rearing practices; gender





## 1. Introduction

Children's tales are not only evocative in their curious portrayals of escapism and adventure, they also contain hidden treasures of meaningful metaphors and insights with which to play and ponder. Such tales can attempt to blur the boundary between childhood and adulthood, truth and fiction, through enabling a playful positionality from an adult perspective, experimenting and exploring like children, at transitional points for growth and development in practice. Indeed, Bruno Bettelheim observed the potential of such fictional tales in benefitting children's emotional workings-through of powerful emotions (Bettelheim [1975] 1991). He states,

> "In order to master the psychological problems of growing up- overcoming narcissistic disappointments, oedipal dilemmas, sibling rivalries; becoming able to relinquish childhood dependencies, gaining a feeling of selfhood and of self-worth, and a sense of moral obligation . . . spinning out daydreams- ruminating, rearranging and fantasizing about suitable story elements . . . give better direction to his life". (Bettelheim [1975] 1991, pp. 6–7)

In a similar way to children, we consider that scholars and practitioners can benefit from stories as a way to re-imagine and work through complex 'high theory' (Strom 2018). This is utilised in taking the stance suggested by Warner in that to 'wonder' and be 'curious' about children's tales can 'unlock social and public possibilities' (Warner 1995, p. XVI).

We thus argue that a 'tale' or 'story' can also hold vast potential in supporting the grasping of difficult, dense and abstract ideas. Such ideas can provide a useful platform from which to explore complex concepts, social norms and philosophy, (see Carlyle et al. 2020 and their interrogation and analysis of attachment theory through the tale Peter Pan). In this paper, we take a position of wonder and curiosity in attempting to highlight how reading Mary Poppins through the lens of Deleuzian philosophy might challenge current discourses around gender, parenting and childcare practices. As such, this paper is a provocation and an invitation for re-imagining gendered and developmental trajectories in current notions of parenting and childcare. We have included our own imaginings through images intended to be read in synergy with our provocations. By putting to work three motifs from the Mary Poppins stories—*the spoonful of sugar, the umbrella, and the starling*—we illustrate how the character of Mary can be seen to disrupt and de-familiarise the often prevailing societal normative tropes of childhood which endorse a Dionysian view of the child as in need of regulation and control. In addition, we offer (Deleuze and Guattari [1987] 2013) lexicon of 'becoming-woman' as a feasible concept with which to consider alternative ways of being and becoming with regard to parenting practices. Similar to many of Deleuze and Guattari's concepts, 'becoming-woman' is an essential process towards their idea of identity and its multiplicity with 'becoming' being a liberating act and deconstruction of the masculine–feminine binary and as a means to address this schism between male and female representations. This does not mean that one must become physically a woman or imitate a woman in 'becoming-woman'. It is not a mimesis. It is a view of the 'self' (in the loosest sense) as having many parts. We have previously found it useful to draw parallels here with evolution and Deleuze and Guattari in their use of the orchid and the wasp in explicating a symbiotic relationship at the heart of their concept. Thus, the orchid needs a specific wasp to pollinate, and the wasp a specific orchid to gain nourishment from and neither are therefore independent of one another. In this example, what is fundamental is the sense that Deleuze and Guattari suggest the analogy of wasp 'becoming-orchid' and the orchid 'becoming-wasp' is applicable to all aspects of reality and is therefore relational. Hence, there is no woman–man binary, only a process of destabilisation of identity and a simulation from an assemblage of different parts—a multiplicity.

Deleuze and Guattari's various works provide critical explorations of capitalism, consumerism and popular culture. This raises an important question about how the authors would have responded to a reading of their work through the lens of a children's book, particularly one that was itself intensively commodified via film, products, and multiple spin-off projects. Deleuze in particular has been dismissive of popular culture, and whilst it is impossible to anticipate what the authors would have made of this article, his take on the creative limitations of pop videos is telling:

> ' . . . they could have become a really interesting new field of cinematic activity . . . but were immediately taken over by organized mindlessness. Aesthetics can't be divorced from these complementary questions of cretinization and cerebralization'. (Deleuze 1972, p. 60)

What is important here, however, and relevant to our project, is the Deleuzian idea that concepts and ideas should be active and put to work. Theories become a 'toolbox' (Deleuze 1972, p. 2); instruments to be utilised and remixed in order to (re)invigorate our familiar engagements with the world. Drawing on Deleuzian ideas in conjunction with the character of Mary Poppins thus offers potential for new interpretations of childhood behaviours, and demonstrates how children's literature can provide different perspectives which challenge hegemonic understandings.

Deleuze and Guattari's theories are often seen as inaccessible and exclusionary, and this reputation of 'high theory' can form a barrier in times that call for radical and trans-

formative thinking (Strom 2018). Using familiar characters and pop-culture motifs, whilst being a practice they may have resisted, can help us to reflect and think their ideas through, as we resist the separation of theory and practice. Further to this, the artistic creations accompanying this article work to de-territorialise established views of the story and de-familiarise the common Disney tropes in a way that the authors would perhaps have welcomed.

Strom (2018, p. 112) discusses the exclusionary role of high theory, and the barrier of inaccessible language that can lead to feelings of intellectual inferiority. As she states: ' . . . we must commit ourselves to finding ways to interrupting those discourses and other exclusionary mechanisms that keep 'high theory' off limits for all but a select elite few.'

## 2. Deleuze's Child(ren) and 'Grown-Ups'

The perceptive work of Hickey-Moody (2013) in her reflections around 'figuring childhood' is a useful place with which to begin expanding further understanding of how Deleuzian philosophy can add to alternative notions of identity, and thus scrutinise the practices of parenting and child care in Western societies. For if we are to embrace a notion of childhood and indeed children as generative figures, vectors of affect, always becoming through actual (present experience) and virtual ideas (experience yet to come), then the same should invariably be said of adulthood (Carlyle 2018). Indeed, considering adulthood as a continuum of further 'becomings' is one in which we could amplify adult blocks as meaningful, collective subjectivities (like child blocks) which zigzag across time. Importantly, for Deleuze child-adult states exist side by side. Hence, avoiding codification, binarism and delineation is at the core of his thesis. These are helpful possibilities when considering Western ideas of parenting that differ and diverge from this to those that endorse Dionysian (lacking in discipline) ideals and values, as opposed to Apollonian (having reason and restraint). The dominance of such ideals permeates a panoply of parenting programmes within the UK, with behavioural approaches and behaviour modification at the heart of many manualised interventions. The 'Incredible Years 'parenting programmes developed successfully by the renowned psychologist Carolyn Webster-Stratton combines behavioural and social learning theories (Webster-Stratton 1992). Although its emphasis is on strengthening the parent–child relationship, it does so in a way which Deleuze's child is seen as 'territorialised' (regulated and controlled). In point of fact, other behavioural programmes such as the 'Triple P' parenting programme use ideas that children should have 'boundaries and limits' set around (mis)behaviour, in 'assertive discipline' (Sanders 2008, p. 509). Whilst this usefully has children's wellbeing at its core, and intends to support parents in understanding child development, it runs counter to Deleuze's concept of children who make their own 'maps' with Deleuze asserting how such 'maps of these trajectories are essential to psychic activity' (Deleuze 1998, p. 61). We therefore could be seen to inadvertently stifle children's creativity, flourishing and zest for life through such limitations. If child development is to be only seen through a medical-psychological discourse, then it is both troubling and constricting. Nikolas Rose and his critique of the 'psychologists gaze' offers a salutary warning about such restricting forces and the manifestation of complex 'apparatus' targeted at children such as the welfare system, the school and the surveillance of parents (Rose 1999). Furthermore, despite the idea of children having 'agency' and them having a 'voice' being afforded much scrutiny in childhood studies (James and James 2004) its translation into policy and practice could be argued to remain somewhat marginal (Aynsley-Green 2018). What seems tantamount to a 'technical normalisation' of childhood through scales, measures and developmental milestones, childhood and parenthood have become governed, and in particular a parent can be are considered as 'a technician in child guidance' (Rose 1999, p. 182). This has also been critiqued as the 'professionalisation of parents' and the fact that in the UK and mainland Europe there is a policy emphasis on child rearing and need for parenting support and programmes (Ramaekers and Suissa 2012, p. 23).

Deleuze's philosophy is one of practical application, encompassing movements, affects, forces and intensities in the multiple subjectivities incumbent in the kinesthetic and sensorial nature of what constitutes a life. It dis-engages from the dominant representations of femininity and masculinity, disrupting assumptions through radical alternatives of what constitutes a human subject. It calls for a new vision of subjectivity and this is the crux of Deleuze's thesis. Indeed, it calls for what Braidotti aptly terms 'dis-identification' (Braidotti 2013, p. 168). This means that 'dis-identifications' equate to a nomadic theory of post-anthropocentric configurations of identity. Such a shift does indeed test the vivacity of the Humanities. As Braidotti asserts:

"We need to overcome this model and move towards an intensive form of interdisciplinarity, transversality, and boundary-crossings among a range of discourses. This transdisciplinary approach affects the very structure of thought and enacts a rhizomatic embrace of conceptual diversity in scholarship.". (Braidotti 2013, p. 169)

Through consideration of 'life' and seeing it as flows of becoming, with complex heterogeneous assemblages and relations, we can embrace a philosophy that encourages the Humanities to encounter contemporary physics and biology in new creative ways. What we have depicted in Figures 1 and 2 is an evocative Deleuzian sense of how it is representing being in the world through what Merleau-Ponty refers to as a 'desiring body'. In this Merleau-Ponty draws our attention to the intertwining of the subject with the world and others. We incorporate this as an affective, intimate, folded-in experience; we are of the world and are entangled with it moment-to-moment. It is an internal relational process which makes a multiple system of two bodies that 'slip into each other' (Merleau-Ponty [1962] 2012, pp. 369–70). This is encapsulated through the phenomenon of emotional contagion. It emerges from an open, un-boundried body (Manning 2009). Thus, the body is no longer engendered or docile and a tacit dimension (Shotwell 2011). As Verhage (2014) aptly asserts, shared meanings and habits stick us together in a kind of intersubjective dance. As Travers plays with the tacit behaviour of Mary Poppins, one in which she is part of the patriarchal Western culture, we suggest by viewing her social world not through a docile body but one which interjects a particular rhythm opposing such patriarchy whilst 'becoming-with it'—she both repels and replays (enacts) ideas of masculinity and femininity. As Bell Hooks forewarns, this is not a consuming of the other, but must be an egalitarian becoming-with (Hooks 1999). The thorny issue is one of embodied possibilities which ruptures and reconsiders the potential oppressions of patriarchy through an intimate moulding and shaping of one another (Verhage 2014). It is a non-gendering, intimately close and borderless relation, whilst in a parallel process retaining the shape of oneself. We therefore concur with the idea of an emerging subject, rejecting ways in which discursive power produces and performs sex/gender through the structure of a heterosexual matrix (Butler 1990). This means subjectification is not seen as fixed and stable with disparate entities, but as a Deleuzian process of affective forces and flows of material-discursive power. From a feminist perspective, we find Deleuze and Guattari's work appealing as it can trouble the political discourse and dominant order around identities through dissolving sexual differences into 'an inhuman flux' (Colebrook 2002, p. 149). As Braidotti asserts, we must contemplate the idea of 'nomadic subjects', bodies which form multiple identities through various activities (Braidotti 1994, 2003). The idea that one is woman, white, and middle class is not just a question of being but the culmination with activities with other bodies and other political distinctions. We offer a re-thinking of the body beyond the male–female binary. What this means is that Deleuze and Guattari's concept of the 'virtual' (that which is yet to be) must be given greater prominence in the Humanities.

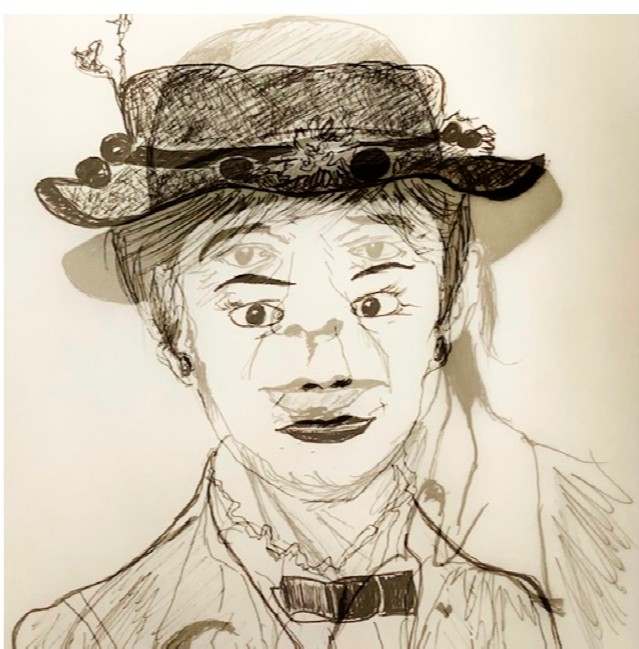

**Figure 1.** Deleuze becoming Poppins transformation: Embodied symbiosis on a molecular level. (All images are original and drawn by Donna Carlyle).

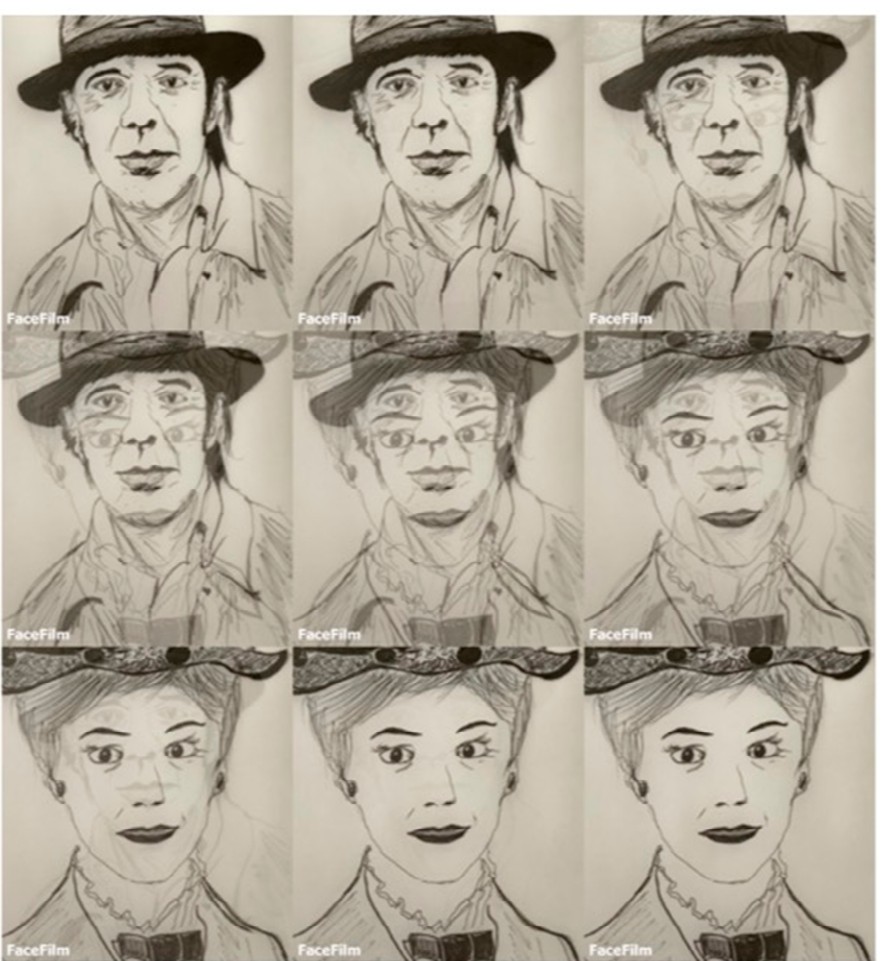

**Figure 2.** Deleuze becoming Poppins: Polymorphous process of virtual possibilities and dis-identification.

### 3. A Spoonful of Sugar Helps the Discipline (Conformity) Go Down

'A spoon was attached to the neck of the bottle, and into this Mary Poppins poured a dark crimson fluid' (Travers [1958] 2018, p. 8).

Children are more attuned to their senses than adults (Bartos 2013). Their sensory prowess is linked to limited life experience as children utilise sensorial ways of knowing, (Hojgaard and Sondergaard 2011). Indeed, Leder (1990) claims that adults are 'disembodied' in attuning less to our sensuous world through modern lifestyles which encompass technological advances in communication, making an 'absent body' evident. It is rather interesting that the film adaptation of Mary Poppins includes the song 'a spoonful of sugar'- a song composed by Robert Sherman in 1964 (Paris 2020). Apparently inspired after his son received his polio vaccine, having been given it on a cube of sugar, he realised its potential and appeal. We suggest that (inadvertently) the song 'A spoonful of sugar' to encapsulate the taking in of something just as unpalatable like the 'medicine' (dark crimson liquid) of scientific parenting manuals' supporting the notion of helping parents to produce a 'good' child (please see Figure 3). Although Travers was inevitably disgruntled with the Disneyfication of her protagonist, the 'spoonful of sugar' is a powerful metaphor for lessening the effect of something unpleasant and unsalutary such as the lack of agency (through a relational ontology) for children. The sweet liberty and de-territorialising impact of the 'spoonful of sugar' leaves a much better taste in the mouth. Despite Travers being generally un-enamoured by the whole adaptation to screen, perhaps our Deleuzian lens can entertain the taste buds in a more playful and productive way. In taking in the parenting (medicine) with sugar ('lines of flight', flowing, leaking and escaping) we can be reassured that the Deleuzian, post-human child can be nurtured and guided in equal measure. This is a kind of parenting that Ramaekers and Suissa (2012) advocate in their work which counter-balances the overly empirical and mainly psychological focus on parenting which is dominant and prevailing.

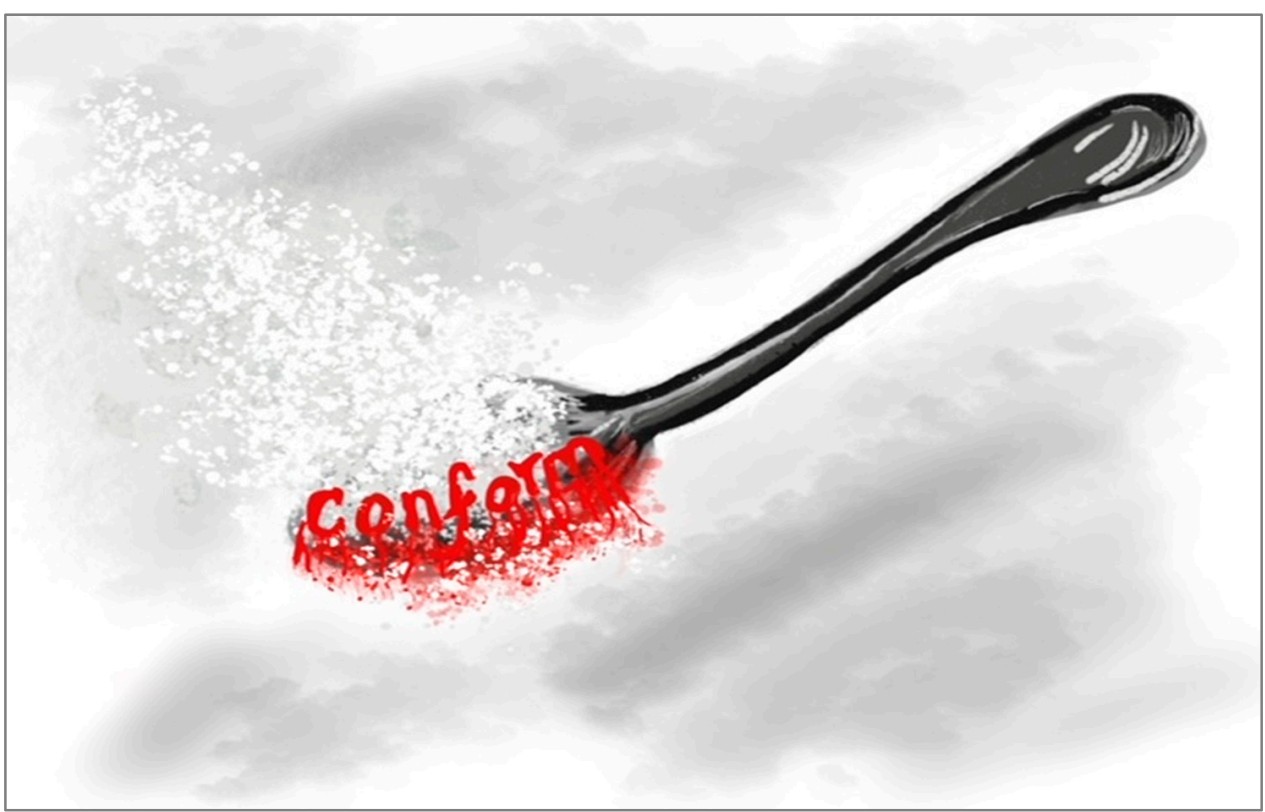

**Figure 3.** A spoonful of sugar". The 'medicine' and scientific-medical dose of 'parenting professionalisation'.

Like Deleuze and Guattari, they propose a philosophical approach to parenting to further enrich the parent–child relationship, placing it in a contemporary space for reflexive thinking about childrearing practices. Travers has characterised Mary Poppins in a very ambiguous manner in that she is frightening and exciting at the same time to the children. Indeed, you could say that she holds a rather 'authoritarian' position and style of parenting as a nanny (Baumrind 2012). However, there is a subtle hint that Travers wished a different flavour to also come through in her protagonist Mary Poppins as she engages in taking the 'medicine' becoming-with the children, 'Mary Poppins then poured out another dose and solemnly took it herself' (Travers [1958] 2018, p. 9). Invariably this foretells a Mary Poppins character that has multiple subjectivities, a free thinking, nomadic spirit that escapes the socially constructed ideas of nannying and parenting, albeit in a blatantly contradictory way (her iconic dress and uniform of a nanny, her regimented temperament). Yet, this is the sublime nature of seeing Travers work not through its Disneyfication but its post-humanism and post-structuralism.

## 4. Umbrellas and Nomad War Machines

In Deleuze and Guattari's (Deleuze and Guattari [1987] 2013) concept of the Nomad War Machine, being 'nomadic' means somehow living outside the current state of affairs. Whilst accepting our embedded, embodied nature, Nomad War Machines (NWMs) are mobile agents who operate in 'smooth' non-hierarchical spaces, not the 'striated' zones of the family or state, where moves are regulated and bureaucratised: 'The nomad intentionally lives without roots; willingly moves from place to place, idea to idea, and concept to concept. Nomads are open to interrelationships of what is before them, even if these interrelationships present places and concepts not traditionally linked' (Clarke and Parsons 2013, p. 9). Mary Poppins acts as an NWM in her transient and fleeting attachment to the Banks family, and her ongoing detachment from the kind of thinking that prioritises loyalty to the hetero-patriarchal structures above loyalty to the self and others. NWMs traverse the boundaries and borders which might constrain them; for Mary this involves processes of de-familiarisation, dis-identification and disruption. She has no work references, no contract of employment and resists any attempt made by the Banks family to constrain her into fixed notions of 'days off.' The notion of Mary as NMW is manifested not only metaphorically, but in the character's physical flight via an umbrella from space to space; she goes where she is needed, rather than where she should be through duty or the constraints of employment relations. Her promise to '... stay til the wind changes' (Travers [1958] 2018, p. 11) thus troubles the normed and humanistic notions of power relations and responsibilities within child-rearing discourse. Nomad War Machine is a practice that is about movement rather than stagnation and results in the creation of temporary becomings, emergent gatherings of people who group themselves around a project or idea, always for a limited time. The notion is counter-cultural within neo-liberal systems of labour and theorisations of attachment, and in this way Mary Poppins is able to work across physical and disciplinary silos and boundaries, beyond the schooling and family 'order machines' (Krejsler 2016). Hence, we concur with other researchers in their evocation of Deleuze and Guattari's ontology (see Duschinsky et al. 2015) in that attachment should not be seen as a 'Sovereign Power' in being politicised as discourse which 'enslaves' mothers through constructing them as solely responsible for childcare. We agree that attachment should be considered as dynamic forces which embody an 'optimism for living' (Berlant 2011). Patriarchal institutions can be shaken in what could be considered social anarchy and a radical feminist force embodied in Mary Poppins. This is also later materialised through Mrs Banks and her support for the suffragette movement, a submissive woman at home but a social anarchist outside that institution. Deleuze's notion of 'becoming-woman' is as radical as Mary Poppins challenging the social construction of gender in her betwixt position which also contradicts a submissive nanny *and* social anarchist. This aspect of Mary Poppins, to make the 'normal' seem strange, is sublime in its protagonists feminist conformity, as Mary Poppins leaves us (and the children) craving for more social transformation. Mary

Poppins portrays radical political activism as its seemingly lingering message through a Deleuze-Guttarian lens, in a sense highlighting Zizek's notion of working with the state (becoming-nanny) to effectively become-revolutionary (Zizek 2007). There are few fictional heroines such as Mary Poppins in such a radical betwixt position.

Re-imagining Butler's (1990) work on 'performativity' and how we are socialised to perform gender roles can be juxtaposed with what we consider a Deleuze and Guattrai 'social anarchist' positioning. We do not suggest Mary is a 'man' to start with, (as in Deleuze and Guattari's assertion man must first become woman) but how 'man'-kind can embrace both female and male characteristics and that binary differences create demarcations and stifle alternative becoming's through how we might be socialised in society. Indeed, in being anti-Oedipus Deleuze and Guattari were against the patriarchal notion of the family (as originally purported by Freud to be one of the core concepts, the Oedipal complex, of human development), (Deleuze and Guattari [1983] 2000). We agree in the sense that both Deleuze and Guattari and Mary Poppins are 'social anarchists' who can support a greater reflexivity when we consider child-rearing practices and gender roles, as most primary care givers are still female (Smith et al. 2021).

Deleuze and Guattari argue that 'a society is defined by its lines of flight...there is always something that flows or flees, that escapes the binary organisations, the resonance apparatus, the over coding machine.' (Deleuze and Guattari 2005, p. 216). At the end of the novel, Mary Poppins takes a literal line of flight, via her umbrella, from the family unit and bounded subject position as Nanny, carer, and from attachment-as-usual. By connecting to natural 'potentia' energies (that is, affirmative, fluid and generative forms of natural power as opposed to hierarchical, organisational 'potestas' power (Braidotti 2018)) Mary extends notions of care and duty beyond the human subject: '... she smiled as though she and the wind understood each other' (Travers [1958] 2018, p. 156).

In another example of Mary's radical 'radicles' (lines of flight and rootlet spurts of growth), she also co-opts the dog 'Andrew' in her social class warfare, (please see Figure 4) (Andrew is anthropomorphised as Miss Lark's 'Little Lump of Sugar' (little boy) (Travers [1958] 2018, p. 41) who reluctantly leads a rather 'luxurious life' (Travers [1958] 2018, p. 39). Andrew befriends Willoughby, a stray 'common' dog, half Airedale and half Retriever. In a defiant act of unity with his new friend Andrew insists that Willoughby should live with him and Miss Lark, and in an enchanting exchange of words between Mary, Andrew and Miss Lark, in which Mary 'speaks dog' (translating this for Miss Lark) Miss Lark is forced to relent or lose her 'Little Lump of Sugar', (please see Figure 5). The matter-of-fact nature of Mary's communication with both dog and adult reinforces the sense of the non-human world being open and responsive (should we only wish to hear it), with all actors being part of a community in constant intra-action and state of exchange. Animals here are thus interacted with not in an anthropomorphising way, but as a move of humility, kinship and care.

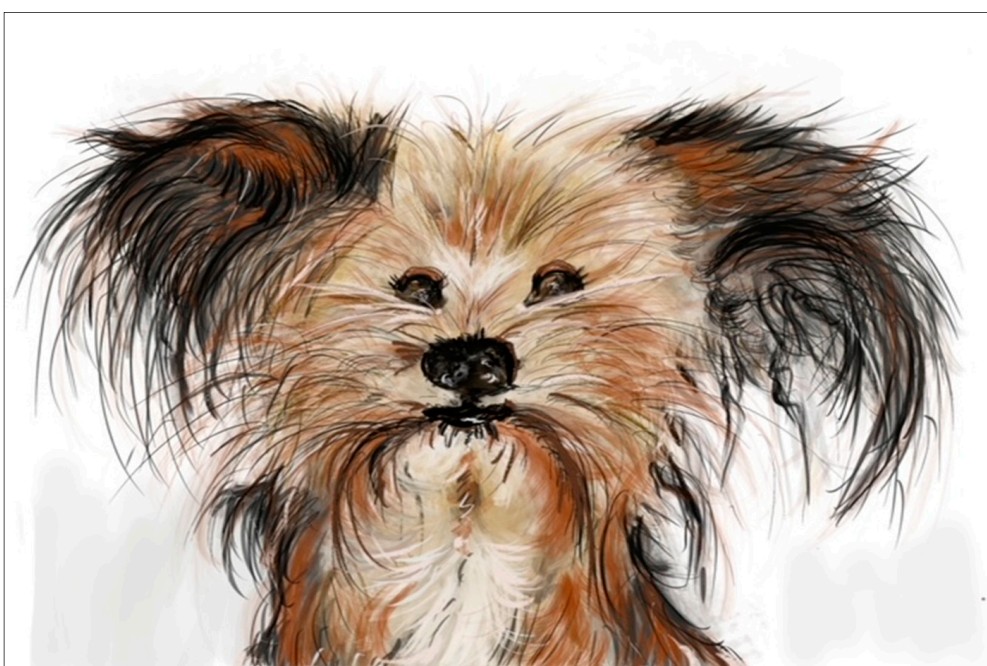

**Figure 4.** Mary 'becoming-with dog' by co-opting 'Andrew' in her social class nomadic war(fare)machine (NWM).

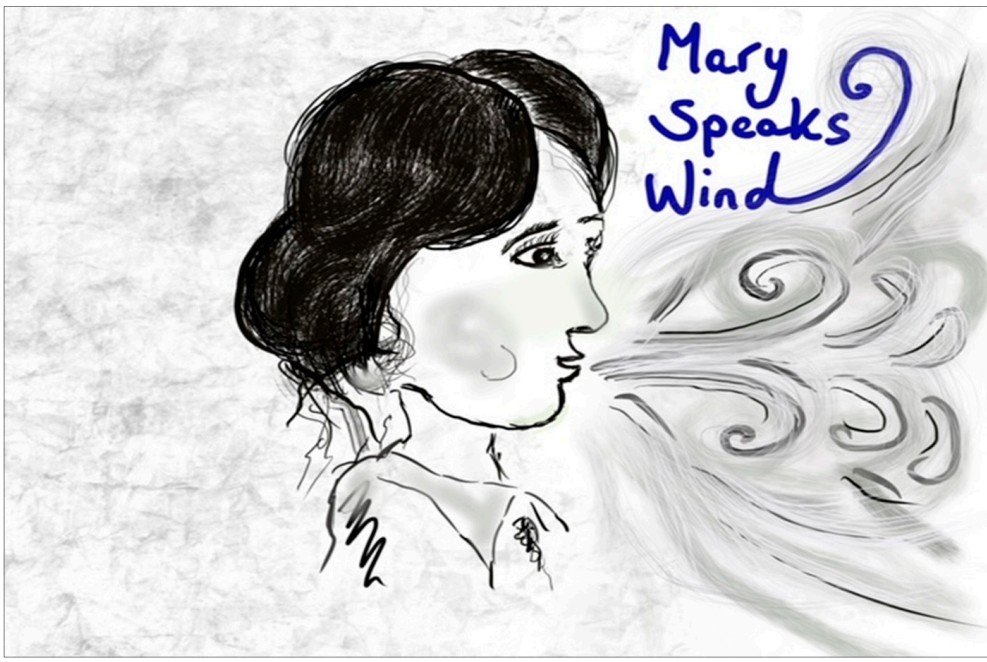

**Figure 5.** Mary becoming-post-human, she speaks wind, bird (starling), dog and sunlight.

## 5. Becoming-With the Starling

The adult–child binary is continually reinforced in contemporary schooling through a distancing process in which adults deny their own childhoods and children are not recognised as taking an active part in knowledge construction (Matthews 1994). Scientific and developmental approaches to childhood similarly position children as 'not fully-formed humans' (Murris 2018, p. 80), imbricated in a linear process of maturity towards the ultimate goal of attaining adulthood, and thus full humanity. This process, reflected in the 'ages and stages' approach of Piaget and other developmental psychologists, echoes recapitulation theory in which the child moves in fixed stages from 'savage' towards

civilisation (Kromidas 2019). The figure of Mary Poppins troubles this one-directional notion of knowledge and skills accumulation via the 'conversations' between John and Barbara, baby twins, and a starling who regularly visits their nursery. Although without speech, the babies are able to converse freely with Mary, the bird, the wind, and sunlight. On discovering that soon they will undergo the 'Great Forgetting', in which they lose the ability to talk to non-human others, the children are distraught. Maturity here is thus seen not as something to be gained, but as *lack*, as receptiveness to worldly-knowing is shut down and thinking becomes hardened and fossilised. As Mary states: 'You'll hear alright...but you won't understand' (Travers [1958] 2018, p. 111). On returning shortly after the twins' first birthday the starling realises he is no longer able to engage them in conversation; unlike Mary (who the bird refers to as 'the Great Exception') the children have matured as predicted and lost the ability to communicate with non-humans, (please see Figure 6).

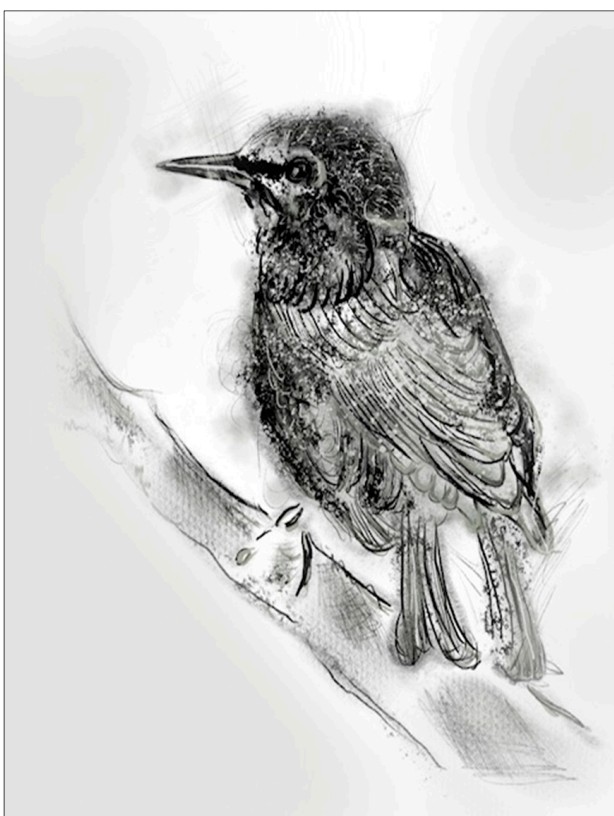

**Figure 6.** The starling is saddened when the children undergo the 'Great Forgetting' and refers to Mary Poppins as the 'Great Exception'.

Despite the various departures or 'lines of flight' taken here which disrupt and subvert normal understandings of childhood, Mary acknowledges that there will always already be a process of return to the familiar hegemonic ways of being-child. This 're-territorialisation', in which a fugitive move is recaptured by normative ways of being, is acknowledged by Mary in a matter-of-fact manner ('Well it can't be helped. It's just how things happen'). However, the implication that she will continue bearing witness to, and facilitating, the ways in which pre-verbal children communicate with the natural world is a hopeful one. Whilst she does not attempt to change the status quo of progress through childhood, Mary's presence and the inversion of the existing order invokes a different kind of sensibility and connection with the world. The children's communication with non-humans and natural forces is not imaginary but positioned as another way of knowing and being; in this way it de-familiarises the reader as we disengage from dominant normative understandings of linear child development and adulthood viewed as the ultimate stage in maturity. This form

of 'conceptual disobedience' (Braidotti 2019, p. 140) positions the agency of young children outside the bounded constraints of the family unit, and Mary as facilitator and interlocutor rather than parental or teacherly figure. In this way she is not only the disruptor of social codes but a conduit for more response-able ways of relating to the world: ' . . . beneath this [inversion of power] there is a less conscious level that gestures towards the contents of the collective unconscious, where things remain unexplained, elusive, but suggestive and evocative' (Perez Valverde 2009, p. 268).

In re-thinking 'bodies', Deleuze and Guattari offer an appreciation of the notion of a lived body on both an immanent and transcendent plane, which in turn is indispensable to an appreciation of the concept of becoming. Becoming-woman, we argue, is a challenge to the dualism enforced by gender binarism as a way to consider how we should no longer determine a body's experience along gendered developmental trajectories. Deleuze and Guattari's becoming-woman is a particularly viable concept that contributes to historical and contemporary discourse and popular culture (as in our textual analysis with Mary Poppins and Travers) to transcend misunderstandings of traditional oppositions and polarities between ascribed and performed masculine/feminine identities. We argue for 'potentia' and potentialities of bodies to become other than that to which they have been assigned. Becoming-woman is not an attempt to become the physical embodiment of a 'female' but to conceive of the idea that the human body is a multiplicity of forces. This is what Mary Poppins portrays, a multiplicity of forces to which we could ascribe both feminine and masculine traits. Her body is attuned to human and non-human bodies, relational forces which are evocative of her body not being contained within the borders to which her sex ascribes. She is never complete and never predictable, aligning with Deleuze and Guattari's notion of transitions or *becomings.* She offers an untold re-working of the body as a reasoned, unified and organised organism, changing it to one in which the body is decoded, disorganised, and de-territorialised.

## 6. Conclusions

Whilst Gilles Deleuze and Mary Poppins at first appear an unlikely pairing, what resonates in this diffractive reading is their shared positionality on the vicissitudes of childhood. They also embrace the notion of transformation at an ethical and political stratum. We argue that this is an ongoing challenge in everyday practice but one which we all should ponder. We can then open up new 'potentia' for growth and flourishing, not only in the children we care for, but ourselves. At times, the post-structuralist ways of becoming seem rather esoteric and intangible; however, when read alongside Travers' subversive tales of caring differently the texts enable an alternative and workable post-human perspective which makes tangible the vast and complex realms of childhood experiences. The words of Zackin (2013, p. 15) encapsulate the author's own desire for a new kind of humanity:

"Mary Poppins views the world from a holistic lens in which everyone and everything is connected."

We hope to evoke a view of this connected world from the author's multiple perspectives; however, as Travers also encourages multiplicity and authenticity of individual experience through Mary's claim: "Don't you know that everybody's got a Fairyland of their own?" (Travers 1982, p. 218). Both Deleuze and Travers-Mary-Poppins thus compel us to explore other modes of being and experience. We invite scholars to take up this same mantle and imagine how the transformative notion of becoming-woman and 'womanly love' is offered in other unexpected places to counter the 'difficulties surrounding the concept of hegemonic masculinity' (del Saz-Rubio 2019, p. 216). In collapsing the feminine–masculine boundaries, all others are altered at both micro and macro levels. Mr. Banks is testament to this, as demonstrated through the transformations in his relationship with his children; seeing them, hearing them and ultimately growing and becoming-with them, becoming-woman. His hierarchical, authoritarian model of parenting and father–child relationship is challenged and successfully opposed by Mary Poppins. Invariably the very identity of Mary Poppins as a nanny is something of an oxymoron and we suggest she is

not a nanny but a nomadic, nurturing desiring-machine, one which desires a new kind of humanity. If such a betwixt position can be re-imagined across multi-disciplinary professions then there is potential to enable childhoods to become flourishing spaces through our shared dis-identities (Hackett et al. 2015). This landscape is one in which there is an understanding of diverse human experiences, beyond the current co-ordinates and the deployment of the 'professional gaze', through parenting interventions and treatment programmes for those who 'fall within or are placed outside of their boundaries' (Ball and Collet-Sabe 2021, p. 7).

As Travers states in a remarkably Deleuzian tone, she 'never wrote for children' and 'if you are honest—you have, in fact, no idea where childhood ends, and maturity begins. It is all endless and all one). And from time to time, without intention or invention, this whole body of stuff, each part constantly cross-fertilizing every other, sends up- what is the right word?- imitations' (Travers 1999, pp. 182–83).

Walt Disney was rather good at 'animating personalities' and creating cartoon icons with the inimitable Mickey Mouse gaining 'persona' and being elevated to stardom (McGowan 2019). What is intriguing is that Mary Poppins was both animated and embodied in an actor, namely Julie Andrews. Travers' discontent might have sprung from her distaste for Disney's 'stardom' and the commodification of Mary Poppins, but nonetheless, their alliance did afford a stage for exploring and discovering new, alternative ideas of childhood and parenting practices, whether directly or indirectly intended. Using the character of Mary Poppins (and popular culture generally) is a means to access Deleuzian ideas that allow us to dis-identify with and trouble hegemonic ideas of child development and parenting-as-usual. We consider that this opportunity to create new concepts both for, and with our times can be *supercalifragilisticexpialidocious*! Now, quintessentially, the idea and the origin of that neologism herald another complex and fascinating tale.

**Author Contributions:** Writing—original draft preparation: D.C., and K.S.; writing—review and editing: D.C. and K.S. All authors have read and agreed to the published version of the manuscript.

**Funding:** This research received no external funding.

**Institutional Review Board Statement:** Not applicable.

**Informed Consent Statement:** Not applicable.

**Acknowledgments:** The authors would like to thank the reviewers for their thoughtful comments.

**Conflicts of Interest:** The authors declare no conflict of interest.

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
