# Peer review of "Deleuze Becoming-Mary Poppins: Re-Imagining the Concept of Becoming-Woman and Its Potential for Challenging Current Notions of Parenting, Gender and Childhood"

_humanities, doi:10.3390/h10040113_

Round 1
Reviewer 1 Report
I am largely sympathetic to this article in which it is attempted to read Travers’ literary work through a Deleuzian lens. There are two minor points I would like to address, though, and I have one major issue, or rather, big question.
First minor point: I think the opening paragraph on Bettelheim is a bit odd in this context. To be clear, I have no problems with it, and if the author would like to keep it, that’s fine with me, but this paragraph stands a bit apart from the rest of the article. If the overall framework is specifically Deleuzian, this opening paragraph is quite general and it is never returned to in the remainder of the article.
Second minor point: this text is only a draft, but I observed quite a number of typos (esential, 90 / polcy 98 / partiacrchy, 135 / dispraite, 144 / re-thinking (of is missing), 152 / compsed, 172 / fortells, 198 / Giles, 310), and perhaps this list is not exhaustive. So, a meticulous copyediting is required.
Major point: Deleuze and Mary Poppins are an ‘unlikely pairing’ (line 310). It is good that the author acknowledges this as the opening sentence of the conclusion. But nonetheless I could not get out of my head that by reading MP through a Deleuzian lens Travers’ work is perhaps given (a bit) too much credit. The article gives the impression that Mary Poppins is an experimental and radical work / character (she is not a ‘nanny but a nomadic, nurturing desiring-machine’, with characteristics of post-humanism). Insofar she belongs to popular culture, the article implies, this seems predominantly due to the film adaptation that led to a Disneyfication of MP. Is it really the case that the radicality of Travers’ text is very much underestimated, and that the text is truly as subversive (with its focus on dis-identification) as the author wants to make us believe?
I myself am most familiar with Deleuze’s two books on cinema, and the second one on the time-image is primarily devoted to quite radical cinema on a formal level (Resnais, Robbe-Grillet, Godard, and so on). It is not unjustified to read popular work against the grain (quite some scholars put this into practice, think only of Slavoj Žižek), but to what extent is it in the spirit of Deleuze’s philosophy to read a relatively popular work (rather than an avant-garde / experimental text) through a Deleuzian lens, as if it is a truly radical work. At the risk of redundancy, I do not think that this project lacks justification, but I would be happy if the author were to address this issue more explicitly.
And thus my main question is: does MP really have the radical potential as suggested here, considering that the text is not really innovative on a formal level? To be honest, I have never read Travers’ text, but given its popularity I do not think it is really radical on a formal level. I do not know whether the comparison is really justified, but I am thinking of Pippi Longstocking. She is a totally unorthodox character, very idiosyncratic, but if she were really as subversive as she seems to be at first sight, she could not have become so popular. She owes her popularity to the way she defies authorities, but these authorities deserve defiance because their attitudes are too strict. To sum up, I raise these points, because I think that Deleuze would have attributed radical potential to works that are challenging in both content and form (Godard, e.g.), but is MP that radical in terms of form that it warrants such an approach. I think that the article would benefit if it were to reflect upon this issue. In the current draft, the fact that Deleuze and MP are an unlikely pairing is basically mentioned in passing, but I would encourage the author to further reflect upon that statement.
Author Response
Response from authors:
Thank you kindly for your time, valuable comments and insights in reviewing our paper. Our points in response potentially encompass and strengthen the helpful critique you gave. It is very appreciated. Please also see highlighted revisions in yellow in the re-submitted manuscript.
- We have addressed the typos. Several were also in relation to the American versions of the word spelling.
- We have re-worded the opening paragraph referring to Bettelheim and contextualised this more appropriately in conjunction with our aim and argument (L 36-39)
- We have emphasised how our paper/project is a provocation and invitation for re-imaging different perspectives (through synergy of text and images/figures) (46-51)
(a) We have strengthened and enhanced our argument for our project (in the introduction) by expanding upon how Deleuze’s theories can offer a ‘toolbox’ and potential for new interpretations of childhood behaviours and we demonstrate how children’s literature can provide different perspectives which challenge hegemonic understanding (L84-89)
(b) the above forms part of a wider expansion and explanation of our argument, added in the introduction (L 71-103)
- In response to ‘content’ and ‘form’ (of Mary Poppins) we have enhanced our discussion through expansion of the concept of ‘becoming-woman’ (L196-214)
In addition, we have a greater inclusion and integration of this Deleuze and Guattari concept in synergy with our aim of being evocative and inviting ‘re-imaging’s’ of both authors as they relate to historical and contemporary practices. Our synergy between the text and image is to provoke and invite these alternative ways of considering growth and develop (in children and in academia) through the lenses of posthumanism and post-structuralism.
- Related to point 5 – we have developed the idea of Mary Poppins and her inimitable betwixt position, including reference to Zizek’s work (which you kindly alerted us to) (L 298-310)
- Reiterating Judith Butlers’ work on ‘performativity’ and performing gender are juxtaposed with a Deleuze and Guattari social anarchist positioning, incorporating their anti-Oedipal stance to enhance and strengthen the argument they make around patriarchal notions of family (L 312-323)
- In our conclusion we have added and emphasised the potential of what our project means in terms of change, growth, and transformation in practice.
Reviewer 2 Report
The topic of the theoretical study is very original. It links poststructuralist philosophy (Deleuze and Guattari) and literature. At first glance, unrelated topics were able to combine the authors and find common aspects. I see in this the importance of study. The highest benefit is its originality. The study offers a glimpse into a new reading of Travers' story with the character of Mary Poppins. The analysis that is offered from the point of view of gender is performed from the point of view of postmodern philosophy as well as literary science.
Mary Poppins is a distinctive character, present in literary, musical and film works. It also appears in animated series. We consider its connection with philosophy to be an original contribution and we highly appreciate it. Combining fantasy literature and philosophy in one study is relatively rare. There are several quality studies of Mary Poppins in the scientific literature, but few associate it with philosophy. We have no reservations about the content, as the study is in terms of theory, arguments and division of the text. The article is adequately referenced. There are enough citations, they correspond to the topic of the article and the authors also use citations not older than three years. The study shows all the formal features of an article of academic provenance.
If the authors would like, but I do not make it a condition, I would like to draw attention to the study: Lisa Zunshine, What Mary Poppins Knew: Theory of Mind, Children's Literature, History. Narrative 27 (1), 2019, which they can still incorporate into the article.
Author Response
Thank you kindly for your time, valuable comments and insights in reviewing our paper. Our points are in the main, in response to Reviewer 1 and consider it potentially encompasses and strengthens the helpful critique you gave. It is very appreciated.
- We have enhanced our discussion through the ‘interpolation’ of childcare material is in synergy with the overall argument around childcare as a feminine domain, in which we have strengthen our argument by:
- Reiterating that Judith Butlers’ work on ‘performativity’ and performing gender are juxtaposed with a Deleuze and Guattari social anarchist positioning in which we do not suggest Mary is a ‘man’ to start with (point made by editor) but how ‘man’-kind does embrace both female and male characteristics and the very binary differences create demarcations and different ‘becoming’s’ through how we might be socialised in society.
- We draw attention to how Deleuze and Guattari, in being anti-Oedipus, were against the patriarchal notion of the family (as originally purported by Freud to be one of the core concepts, the Oedipal complex, of human development). We agree in the sense that both Deleuze and Guattari and Mary Poppins are ‘social anarchists’ that can support a greater reflexivity when we consider child rearing practices and gender roles (most primary care givers are still female, Smith et al 2021).